# Developing novel antimicrobials by combining cancer chemotherapeutics with bacterial DNA repair inhibitors

**Lorenzo Bernacchia, Arya Gupta, Antoine Paris, Alexandra A. Moores, Neil M. Kad** \* 

School of Biosciences, University of Kent, Canterbury, Kent, United Kingdom

\* n.kad@kent.ac.uk

**Data Availability Statement:** All data in this study will be available here: https://kar.kent.ac.uk/103993/.

## Abstract

Cancer chemotherapeutics kill rapidly dividing cells, which includes cells of the immune system. The resulting neutropenia predisposes patients to infection, which delays treatment and is a major cause of morbidity and mortality. To tackle this problem, we have isolated several compounds that inhibit bacterial DNA repair, alone they are non-toxic, however in combination with DNA damaging anti-cancer drugs, they prevent bacterial growth. These compounds were identified through screening of an FDA-approved drug library in the presence of the anti-cancer compound cisplatin. Using a series of triage tests, the screen was reduced to a handful of drugs that were tested for specific activity against bacterial nucleotide excision DNA repair (NER). Five compounds emerged, of which three possess promising antimicrobial properties including cell penetrance, and the ability to block replication in a multi-drug resistant clinically relevant *E. coli* strain. This study suggests that targeting NER could offer a new therapeutic approach tailor-made for infections in cancer patients, by combining cancer chemotherapy with an adjuvant that targets DNA repair.

## Author summary

As the number of antimicrobial resistant bacteria are rising it is imperative to find new solutions to combat this existential threat. Patients undergoing cancer chemotherapy are particularly vulnerable to infection due to the damage the immune system receives as a consequence of chemotherapy. Our focus was to hijack cancer chemotherapy to kill the invading bacterial cells by compromising their defences. In particular, we focused on inhibiting DNA repair mechanisms, these are employed by bacteria when threatened by numerous anti-cancer therapies. To do this, we screened ~3000 approved drugs for an effect on DNA repair. Of these, three were able to kill bacteria only in the presence of the anti-cancer drugs, and also showed effectiveness against a multi-drug resistant bacterial strain found in patients with sepsis. These results offer a promising new approach to the design of antimicrobials tailored for use with cancer patients, and potentially an avenue for the development of a new class of antimicrobial.

**Funding:** This study was supported by the Biotechnology and Biological Sciences Research Council [grant numbers BB/P00847X/1, BB/M019144/1, BB/T017767/1] to N.M.K. https://www.ukri.org/councils/bbsrc/. Cancer Research U.K. [grant number A30456] to N.M.K. https://www.cancerresearchuk.org/. MRC-IAA [grant number W596141] to N.M.K. https://www.ukri.org/councils/mrc/. No funders played any role in the study design, data collection and analysis, decision to publish, or preparation of the manuscript.

**Competing interests:** The authors have declared that no competing interests exist.

## Introduction

The fundamental therapeutic approach for cancer chemotherapy is to target rapidly dividing cells by virtue of their need to replicate DNA [1]. However, this approach causes neutropenia by off-target killing of circulating immune cells [2]. Coupled with the chemotherapy-induced degradation of physical barriers such as mucous membranes, pathogen penetration is also enhanced [3,4], further contributing to bacterial infection, the second most common cause of death in cancer patients [5].

Anti-cancer compounds function through a variety of mechanisms including damaging cellular DNA. Among these the first platinum based anti-cancer drug, cisplatin (cis-Diaminodichloroplatinum, CIS), was fortuitously discovered because of its ability to inhibit cell division in *Escherichia coli* [6]. The mechanism was found to include stalling replication through the formation of DNA adducts with inter- and intra- strand crosslinks [7]. Cisplatin was subsequently found to similarly inhibit human cell proliferation and therefore was exploited to treat a variety of cancer types [8]. In both bacterial and mammalian cells, cisplatin adducts are repaired by the specific activity of enzymes in the nucleotide excision DNA repair (NER) pathway [9,10]. Here the similarities end, bacterial NER uses fewer enzymes with no biochemical homology to their human counterparts [11]. In bacteria, NER removes a variety of damage types including cisplatin adducts [12] but is primarily deployed to resolve UV-induced DNA damage. It begins with recognition and verification of DNA distorting lesions by $UvrA_2UvrB_2$, followed by recruitment of an endonuclease (UvrC) that nicks the DNA on the same strand either side of the lesion. This damage-containing oligonucleotide is removed by a helicase (UvrD), before DNA pol I restores the correct DNA [13]. Therefore, with impaired NER, bacteria would not be able to repair the damage caused by cisplatin during cancer chemotherapy [9,14].

In this study, we develop a new potential therapeutic approach for cancer patients by targeting bacterial NER to eliminate microbial infection that results as a consequence of cancer chemotherapeutics and hospitalization. Among the available pathogens we chose *E. coli*, known to be responsible for infections in humans [2,15–17] and even for complications in patients receiving platinum therapies [18]. We have identified a series of NER inhibitors screened from a library of FDA-approved compounds against *E. coli* by dosing bacteria with a sub-lethal concentration of cisplatin and looking for compounds that halt bacterial growth only in the presence of cisplatin [16,18]. Inhibition of NER alone does not kill bacteria [14,19], therefore it is likely that compounds target DNA repair, preventing recovery from exposure to the genotoxic anticancer compound. To confirm the mechanism of action for our lead candidates as inhibition of NER, we have further triaged the pool of hits using a series of *in vivo*, *in vitro*, *in silico* and single-molecule assays. These findings represent a new mode for antimicrobial activity as an adjuvant to cisplatin. We anticipate that this new class of inhibitors could be administered directly to patients receiving cisplatin-based cancer chemotherapy, thereby protecting them from chemotherapy-induced bacterial infection. To provide initial evidence that these treatments may be useful in patients, we have successfully verified the effect of a subset of these compounds against EC958, a pathogenic multidrug-resistant clinical isolate of *E. coli*, globally disseminated and among the major pathogens responsible for urosepsis and hospital acquired infections [16,20,21].

This study offers a significant step forward in the battle against co-infection during cancer treatment, which leads to delays in chemotherapy treatment, and directly risks patients' health. In addition, by defining bacterial NER as a new drug target, this opens the door to the development of new adjuvant drugs that work alongside DNA damaging agents, for wider application against multi-drug resistant bacteria.

## Results

### Screening for growth inhibitors in *E. coli*

The screening protocol used in this study is summarized in Fig 1A and is used to identify FDA-approved compounds that prevent growth of *E. coli* only in the presence of cisplatin. Since bacterial NER is the primary mechanism of repair used for UV adduct formation our initial hits are refined by their ability to inhibit growth following UV treatment of bacteria followed by a series of further tests to confirm the mechanism of action as NER inhibition. To ensure that drug efflux is not a barrier for drug action, thereby maximising the number of hits from our screen of compounds, we created a drug efflux pump (*tolC*) knockout strain of *E. coli* MG1655 (MG1655 *ΔtolC*). To understand the role of efflux, all screens were performed in parallel with WT MG1655 and MG1655 *ΔtolC*. The concentration of cisplatin (4 μg/mL or 13.3 μM) used in the screen was just below the minimal inhibitory concentration (MIC) we recently determined for these strains [14], and all FDA-approved compounds were used at 20 μM. Growth inhibition was determined through the colorimetric resazurin assay (Fig 1B), which relies on active metabolism to convert the blue coloured resazurin to pink resorufin [19]. A screen of 2731 compounds revealed 172 potential NER inhibitors.

### Validating NER as the inhibitor target

The above data strongly implies that the final set of inhibitors work in combination with cisplatin to inhibit bacterial growth. However, to confirm the mechanism of action we performed a series of studies directly testing efficacy against NER *in vitro* and *in vivo*.

We screened the initial hits using UV exposure of both our *E. coli* strains at a sub-MIC dosage of 75 J/m$^2$ at 254 nm previously shown to activate NER stalling DNA replication [19] before introducing the reduced panel of compounds in the absence of cisplatin. This second

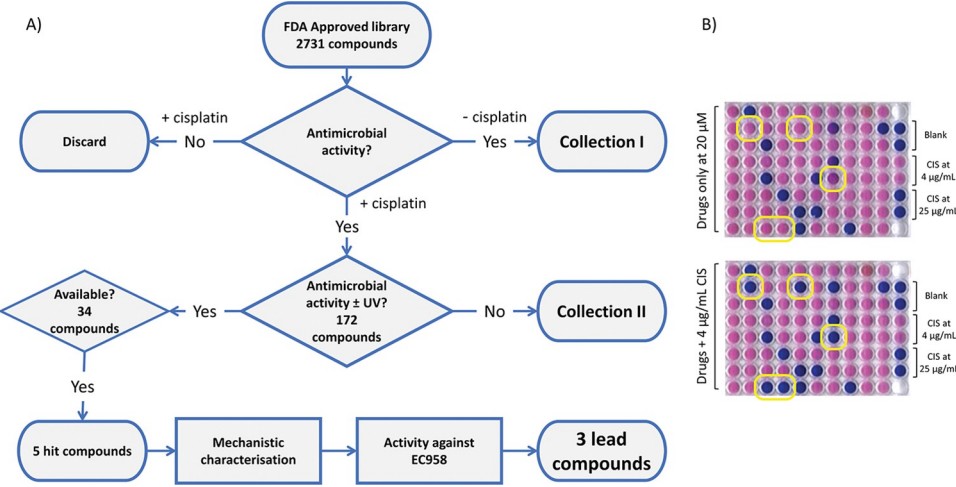

**Fig 1. Screening pipeline.** Phenotypic screening of FDA approved compounds was performed in the presence of cisplatin using *E. coli* strains MG1655 and the efflux pump knock-out MG1655 *ΔtolC*. The latter was used to increase the search area for active compounds. **A)** Shows a schematic of the screening strategy, starting at finding compounds with antimicrobial activity in the presence of cisplatin and then confirming their activity towards NER using a series of mechanistic assays. Activity against the clinical isolate EC958 identified 3 lead compounds from the original 2731. Collections I and II include a number of potential antimicrobials for future exploration. **B)** Growth inhibition assays in the absence (top) and presence (bottom) of cisplatin. These assays use the colour change of resazurin to indicate bacterial growth (pink) or its inhibition (blue). The appearance of new blue wells in the bottom plate indicates drug activity only in the presence of cisplatin. Dual replicate controls are shown on the right lane (top to bottom) for no drug, sub-lethal cisplatin dose with no drug but containing 2.5% DMSO, and a lethal dose of cisplatin.

step resulted in 34 hits (in MG1655 and MG1655 Δ*tolC*, combined). Based on availability, we proceeded to fully characterise the best 5 of these hits (Pirarubicin, Mitoxantrone and 9-aminoacridine active against MG1655 and L-Thyroxine and Dienestrol only active against MG1655 Δ*tolC*). Firstly, we tested for DNA incision in the presence of drug *in vitro*. This crucial step occurs after damage recognition by UvrAB and precedes the resolution aspects of repair and is therefore highly specific for NER. The standard approach for testing incision uses gel-based incision assays [22], however, these are not scalable to high throughput screening and are poorly quantitative. Therefore, we developed a new fluorescence based assay for incision (Fig 2A), which is less prone to photobleaching than another recently developed method [23]. A complementary oligonucleotide pair with a 3' Cy5 on one strand and 5' black-hole quencher (BHQ) on the other is minimally fluorescent. By placing a fluorescein adducted thymidine 14 nt away, but on the same strand as the Cy5, results in an NER-based incision 10 nt from the Cy5-strand end. This leads to the 10 nt fragment leaving the duplex and an increase in fluorescence (Fig 2A). We expressed and purified UvrA, UvrB and UvrC to quantify the incision reaction using this fluorescence-based assay (reporting the data as relative

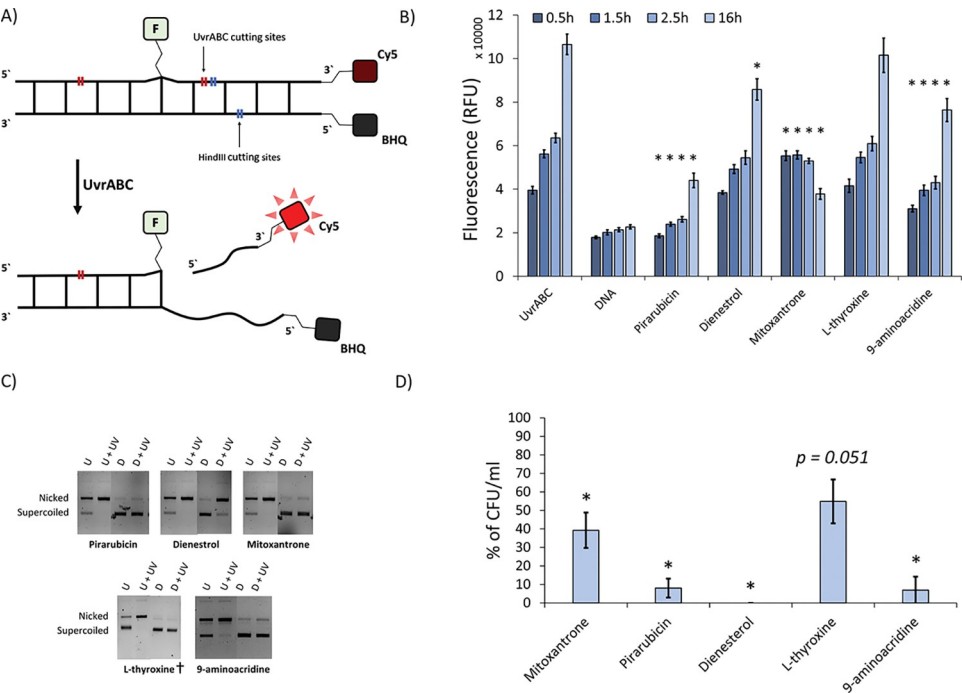

**Fig 2. *In vitro* and *in vivo* tests for nucleotide excision repair inhibition. A)** Schematic representation of the fluorescence incision assay to assess inhibition of NER activity using one oligonucleotide with an engineered damaged site (F = fluorescein) and a reporter (Cy5 = fluorophore); the second complementary oligonucleotide possessed a black hole quencher (BHQ) to quench the Cy5 fluorescence until the top oligonucleotide is nicked by the NER system proteins UvrA, UvrB and UvrC (UvrABC). **B)** Results from the fluorescence incision assay (A) reported as relative fluorescence units (RFU). UvrABC is the control with no drug, and DNA has no drug or UvrABC. The progress was checked at the time points indicated and error bars represent the standard error of the mean. * = p ≤ 0.05 (n ≥ 4 replicates) compared with UvrABC. **(C)** Confirmation of the fluorescence assay with a classical gel-based incision assay demonstrating the inhibition of NER, U is undamaged pUC18, D is the assay in the presence of drug and UV indicates the plasmid is damaged with 200 J/m² UVC (data derives from ≥2 independent replicates) † = ½ incubation time (15 minute). **(D)** Inhibition of plasmid DNA repair *in vivo*. The percent recovery of transformants of pUC18 DNA carrying ampicillin resistance when damaged with 200 J/m² UVC (n = 3) after plating onto ampicillin agar is shown on a scale relative to repair-efficient controls (See S1 and S2 Figs). Although L-thyroxine substantially reduced repair activity, it remained on the borderline of statistical significance (p = 0.051). Error bars represent the standard error of the mean.

fluorescence unit (RFU) Fig 2B), but in parallel confirmed the validity of this approach using a standard gel-based assay (Fig 2C). Pirarubicin (max inhibition of ~97% after 0.5h of incubation), Mitoxantrone (max inhibition of ~82% after 16h of incubation) and 9-aminoacridine (max inhibition of ~49% after 2.5h of incubation) exhibited significant reduction in NER activity, whereas Dienestrol (max inhibition of ~25% after 16h of incubation) showed only partial inhibition of the pathway. L-thyroxine did not show inhibition in the fluorescence assay, but in the gel-based assay, when incubated for a shorter period (15 minutes marked †), did show inhibition (Fig 2C). Pirarubicin and Mitoxantrone are also known to be able to intercalate DNA [24,25], however, other intercalators tested were unable to effectively impair the UvrABC mediated incision (S3 Fig).

To provide a second test that the drugs were targeting NER, we transformed bacteria with a UVC damaged pUC18 plasmid (254 nm at 200 J/m$^2$), carrying the ampicillin selection marker. We reasoned that inhibition of the NER pathway would prevent recovery of transformants on selective agar (S1 Fig shows the results for the *ΔuvrA* transformation control). The results were reported as the percentage of colony forming units (CFU) per ml of solution of the treated samples compared to the untreated bacteria (reported as 100%). As expected, all of the compounds impaired recovery on selective media (Fig 2D) with L-thyroxine at the verge of significance. As a control we also showed that the compounds alone did not impair *E. coli* growth (S2 Fig).

## Drug interactions with the molecules of NER

Having validated that the 5 compounds inhibit NER activity, we investigated how these function at a molecular level. Upon locating damage UvrA hydrolyses ATP, leading to the loading of UvrB [26–29]. We directly measured the rate of ATP turnover for purified UvrA with and without DNA using an *in vitro* NADH-linked assay [19]. When UvrA was incubated with 20 µM of each shortlisted compound, four were found to significantly affect its ATPase (Fig 3A). Among those, Pirarubicin, Mitoxantrone, Dienestrol and L-thyroxine all inhibited the ATPase, with the latter two drugs having the strongest effect. To understand how these molecules bind UvrA, we performed *in silico* docking using AutoDock Vina [30]. ATP was used as a control during complete surface exploration of UvrA; the search successfully located both of UvrA's ATP binding sites based on comparison with crystal structures [31]. The docking precision was good enough to locate specific interactions with residues K37 (proximal ATP site) and K646 (distal ATP site), which have been shown previously as essential for UvrA's ATPase activity [26]. The docking predicted a binding energy for ATP of -9.2 kcal/mol and -9.6 kcal/mol, at the proximal and distal sites, respectively. The higher binding affinity of ATP predicted for the C-terminal site is consistent with a recent kinetic study [32], further validating the approach. The strongest affinities are shown as binding energies for each compound in Fig 3C (further data can be found in S1 Table). Two compounds showed interaction only with the ATP binding sites (Fig 3B); Pirarubicin, which had a stronger affinity than ATP at the proximal site (-9.8 kcal/mol), and Mitoxantrone, which had a preference for the proximal site over the distal, although the binding energy of -7.5 kcal/mol was lower than that of ATP. The docking also revealed two previously unidentified allosteric binding pockets (Fig 3B). Allosteric site 'BP1' bound Dienestrol strongly (-8.7 kcal/mol), and the second allosteric site 'BP2' close to the proximal ATPase cassette bound L-thyroxine (-7.0 kcal/mol) strongly.

To understand if the allosteric binding sites directly affected UvrA binding to DNA we turned to single molecule visualization. Based on our previous data indicating C-terminal fusion of a fluorescent protein to UvrA does not affect its function [27], we constructed and expressed C-terminally fused UvrA-mNeonGreen (UvrA-mNG). Our recent findings showed

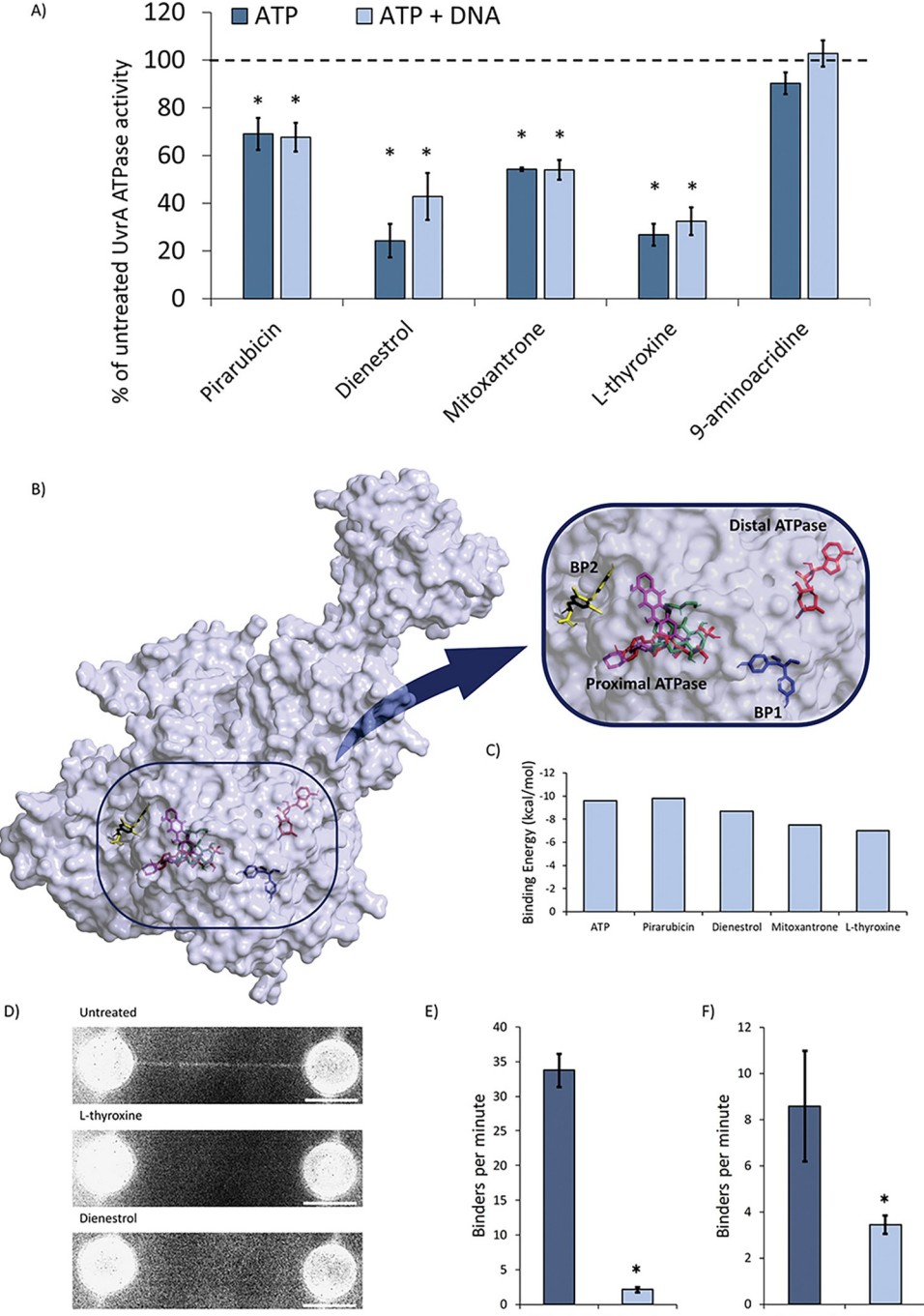

**Fig 3. Inhibition of *E. coli* UvrA binding to DNA by selected hits. A)** NADH-coupled ATPase assay showing the effect of the hits on UvrA's ATPase activity expressed as percentage of that in the absence of drug (dotted line). Both the untreated controls (± DNA) were respectively considered 100% to allow direct comparison with the treated samples. The untreated control showed a 56% increase in catalytic activity in the presence of DNA. The error bars represent the standard error of the mean. Asterisks mark significance: $p \leq 0.05$, n = 3 independent replicates. **B)** The AlphaFold-calculated structure of the *E. coli* UvrA monomer showing the best docking conformation of the compounds with the greatest effect on UvrA's ATPase activity. The zoomed in image clearly shows ATP (red), Pirarubicin (magenta), Dienestrol (blue) Mitoxantrone (green), L-thyroxine (yellow) docking. Remarkably L-thyroxine and Dienestrol bind to previously undetected locations on the surface of UvrA. **C)** The minimum binding energy for each compound reveals a range of affinities, although the absolute understanding of these affinities is not clear the values are close or exceed that of ATP (-9.6 kcal/mol). **D)** Using the C-trap, binding of UvrA-mNG to a single molecule of DNA could be observed. In the absence of any compounds the average combined fluorescence image from

a 10-minute video of DNA shows clear decoration with UvrA (top). In the presence of L-thyroxine (middle) or Dienestrol (bottom) very few molecules bind DNA. **(E & F)** Quantification of DNA binding was provided by the number of binders per minute. This revealed the number of UvrA molecules bound to DNA is significantly reduced in presence L-thyroxine (n = 6 strands, p<0.05) or in the presence of Dienestrol (n = 6 strands, p<0.05).

how UvrA lacks efficient DNA binding activity when in the absence of ATP [27,33]. L-thyroxine and Dienestrol both bind to allosteric sites and have the strongest reduction in ATPase, and have not been identified as DNA intercalators (unlike Mitoxantrone and Pirarubicin [24,25]) making them ideal compounds for single molecule visualization. In this assay, we suspended a single molecule of DNA between two beads caught in optical traps using the Lumicks C-trap system. Using microfluidics, we created two channels one with UvrA alone and the other with UvrA plus drug. The single molecule of DNA was moved between these channels using the laser tweezers. In the absence of drug, UvrA binds well to the DNA (Fig 3D top), however with either 20 μM L-thyroxine or 20 μM Dienestrol we observed a huge reduction in UvrA binding to the DNA (Fig 3D middle and bottom). Quantification of these interactions was achieved by measuring the number of binders per minute over a 10-minute acquisition, this enabled us to demonstrate a statistically significant reduction in binding (Fig 3E and 3F).

## Do NER inhibitors enhance cisplatin cytotoxicity in bacteria?

To study the combined effects of the compounds identified with cisplatin we performed checkerboards assays which are two-dimensional survival assays. For stronger clinical relevance these were performed on the wild-type strain, which restricted the pool of compounds to Pirarubicin, Mitoxantrone and 9-aminoacridine.

A diagram of a checkerboard assay is shown in Fig 4A, the top row shows a serial dilution of the drug to determine the MIC, and the rightmost column corresponds to the serial dilution of cisplatin in the absence of the drug. As the drug concentration is raised (right to left in columns) the MIC for cisplatin drops (yellow arrow), this indicates the drug and cisplatin positively cooperate to inhibit bacterial growth. Each step is a two-fold change in concentration; therefore, the yellow arrow indicates a 16-fold reduction in cisplatin's MIC. Similarly, increasing cisplatin concentration (bottom to top rows) identifies the maximum cooperative effect on drug dosing (green arrow), in the example this corresponds to an 8-fold reduction in MIC. This process was used to determine the highest shift for both compounds able to inhibit growth (Fig 4B). The results were also used to estimate the degree of the cooperation according to the widely used FIC index [34] (Fig 4A). Mitoxantrone, 9-aminoacridine and Pirarubicin all showed 2-fold or greater decrease in MIC. 9-aminoacridine was most potent with an increase in antibacterial activity of 8-fold and increase in cisplatin activity of 4-fold in WT MG1655. The FIC index indicated Mitoxantrone had an additive effect (FICi = 1) with cisplatin, while 9-aminoacridine and Pirarubicin showed partial synergy (FICi = 0.625). Furthermore, we measured the combination of these compounds with the DNA damaging agent 4-NQO [19] to further ensure that the antimicrobial effect observed was due to NER. The results shown in S5 Fig show a clear cooperation of the two agents supporting our previous data.

To move closer to future clinical studies, we also examined if our successful hits were effective against the multidrug-resistant urosepsis-causing *E. coli* clinical isolate, EC958 [35]. Cisplatin's MIC against EC958 was 3.13 μg/ml and 12.5 μg/ml respectively in absence and presence of 2.5% DMSO (S4 Fig). These values are identical to that previously reported for WT MG1655 [14], further supporting the use of this approach against multi-drug resistant bacteria. Remarkably, all three drugs showed enhanced activity with cisplatin: Pirarubicin enhanced the cisplatin MIC 8-fold, 9-aminoacridine 2-fold and Mitoxantrone 4-fold (Fig 4C).

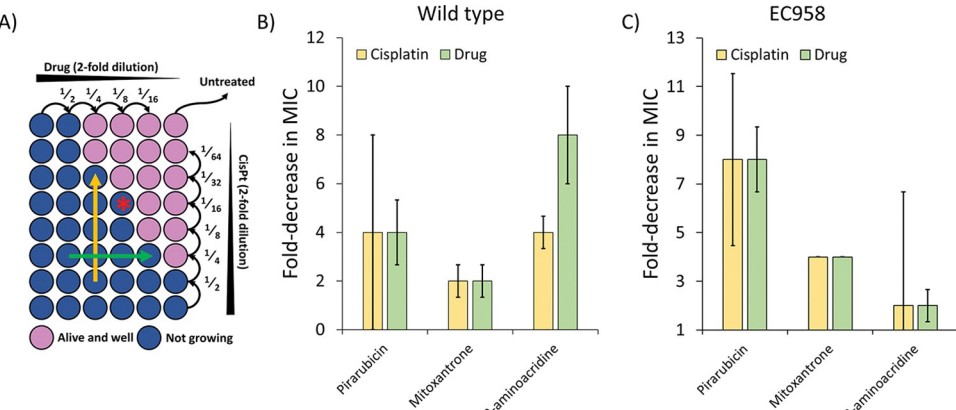

**Fig 4. Inhibitory activity of selected hits against MG1655 and EC958 in the presence and absence of cisplatin. A)**
A representative checkerboard assay plate, drug concentration is decreased left to right and cisplatin decreases bottom
to top. The yellow arrow indicates the greatest decrease in MIC for cisplatin (~16-fold) and for the drug (8-fold) this is
shown as the green arrow, the asterisk represents the well with the most efficient combination (lowest simultaneous
amount for both the drugs). **B)** Bar chart representation of the median fold decrease in MIC for MG1655 when the
drug and cisplatin were combined (as shown by the arrows in panel A). **C)** Same as (B) but for EC958 The panel was
limited to these drugs because of their ability to evade the efflux pump TolC. Data points are derived from three
independent replicates and error bars are the standard error of the mean.

Furthermore, Pirarubicin appeared to have a synergistic effect (FICi = 0.5), whereas Mitoxantrone was partially synergistic (FICi = 0.75) and 9-aminoacridine additive (FICi = 1).

## Discussion

Utilizing compounds that inhibit DNA repair offers a unique approach to tackle infection in immunocompromised cancer chemotherapy patients. This has particular relevance for tackling nosocomial infections in patients that are admitted to hospital. In this study we have developed a new approach to the discovery and testing of adjuvant drugs that possess an enhanced anti-bacterial activity in the presence of the cancer chemotherapy drug, cisplatin. Confirmed using a number of evaluation screens and assays, the adjuvants target nucleotide excision DNA repair. Five compounds were narrowed down from these screens, and among these, three were also shown to possess activity, in combination with cisplatin, against the multi-drug resistant *E. coli* clinical isolate, EC958.

Bacterial nucleotide excision repair represents an ideal target for drug development due to the absence of any structural homology with the human counterparts. For instance, the closest sequence homology between UvrA and any homo sapiens protein is ~30% (blast searching UniProt and Protein Data Bank databases). To our knowledge, there has only been one study that has successfully developed a bacterial NER inhibitor. In that study the investigators used UV to damage bacterial DNA and screened ~40k compounds for effects against mycobacterial NER, [36]. A single effective compound was isolated, however its clinical application is limited because of its poor solubility and potency [19], along with the difficulty of co-administering with UV. Therefore, our approach using cisplatin as the DNA damaging agent and screening for adjuvants from an FDA-approved library offers the potential to rapidly progress to the clinic. The compounds thus discovered are currently used in a number of applications ranging from endocrinology to anti-cancer therefore we don't anticipate effects against eukaryotic proteins different from the already tested activity and therefore we didn't perform tests on human cell lines or proteins. The latter application is not surprising, given the overlap between antimicrobials and antineoplastics has been well established since, both aim to kill rapidly dividing

cells [37,38]. As a consequence, this raises the tantalizing prospect that simply changing the anti-cancer drug treatment regimen might have immediate benefits to patients in terms of reducing nosocomial infection.

## Potential clinical significance of three compounds

Three of the lead compounds were effective against clinical isolates, therefore possessing lead like characteristic and possibly offering the possibility of rapid advancement to the clinic. 9-aminoacridine is currently used as an externally applied antiseptic, however its use as an antineoplastic agent has recently been proposed due to its action on PI3K [39]. Interestingly, 9-aminoacridine has also been used to derivatize cisplatin for improved DNA damaging capabilities [40]. It is therefore possible that 9-aminoacridine functions with cisplatin to severely damage the DNA, which overwhelms NER. This would be consistent with the lack of effect on UvrA's ATPase, however, the clear reduction in incision could equally derive from effects on the other NER proteins. Furthermore, evidences suggest a link between recombination repair and nucleotide excision repair to allow for survival after DNA damage [41]. Therefore, we cannot exclude a 9-aminoacridine toxic effect on other bacterial DNA repair proteins. Mitoxantrone and Pirarubicin are both antineoplastic topoisomerase inhibitors and the mechanism of action for these compounds includes DNA intercalation, although the anthracycline Pirarubicin additionally functions through the generation of reactive oxygen species [42]. It is easy to dismiss the effective drug properties of these compounds because they are mediated through DNA intercalation, however, we demonstrated that both Mitoxantrone and Pirarubicin directly inhibit UvrA's ATPase activity in the presence and absence of DNA. The latter point is extremely important, since inhibition is seen without DNA this indicates that intercalation cannot be the sole mechanism of action. To support this, we also studied a Mitoxantrone analogue (Pixantrone) and known DNA intercalators from the Camptothecin family (S3 Fig); none of these compounds were able to inhibit NER *in vitro*.

## UvrA may have a distinct allosteric binding pocket

Using *in silico* docking we modelled interactions between the compounds and UvrA. Interestingly, Dienestrol and L-thyroxine showed strong affinity for two binding pockets distinct from the ATPase sites. These compounds reduced the ATPase activity of UvrA by >70% in the absence of DNA and ~60% in the presence of DNA, indicating these previously unidentified pockets may act allosterically and offer potentially new druggable targets on UvrA. Furthermore, Dienestrol and L-thyroxine do not intercalate with DNA or inhibit bacterial growth alone, but did impair bacterial cell division when combined with cisplatin or UV radiation. Using single molecule imaging we showed that this was likely due to severely disrupted DNA binding of UvrA. This finding indicates that these compounds can cooperate with these treatments and opens up the possibility of finding more active analogues based on their chemical scaffold.

## Potential considerations of drug administration

Cisplatin is a widely used anticancer agent with a well-established antibacterial activity [6,8,43]. However, the platinum compounds are unable to reach sufficiently high concentrations to exert their antibacterial effects during cancer chemotherapy. The incidence of infection in cancer patients is significantly elevated due to neutropenia and exacerbated by time spent in hospitals leading to nosocomial infection [44,45]. Interestingly, cisplatin has been reported to have significant antibacterial properties, even being able to eradicate persister cells of different species [43], indicating a possible strong bactericidal effect. Therefore, we reasoned

that an adjuvant able to disrupt bacterial NER will lower cisplatin's MIC sufficiently to allow the platinum compound to act as an antibiotic alongside its therapeutic effect against cancer cells. Since the administration of drugs in this study requires the presence of a DNA damaging agent, this limits the period over when the drugs will be active. The pharmacokinetics of both drugs will define the therapeutic window; however, the cooperative nature of the combination means that lower drug concentrations are required to halt bacterial growth, lengthening the interval during which the combination might provide beneficial effects. Furthermore, this pilot study was conducted in defined media and in the presence of DMSO, which has been shown to be detrimental for cisplatin activity [14,46], meaning that it may be more effective *in vivo*. During drug administration in patients cisplatin can reach considerably higher concentrations (up to 75.5 μg.h/ml during a three hour infusion) [47] than required to inhibit bacterial growth, coupled with our adjuvants this will facilitate bacterial cell death. This fixed and very high administered concentration of anticancer drug means that we need only titrate up the concentration of the adjuvant reducing risks from their combination, although future studies will be required to evaluate these risks. At present, we are engaged with further understanding the combined pharmacokinetics and the effects of clinically relevant growth conditions as a precursor for further developments.

In summary, here we have developed a screening strategy to find existing compounds that work in combination with the anticancer therapeutic, cisplatin, which opens up huge potential for the development of new antimicrobials. The compounds we have identified offer a starting point for entry to the cancer clinic, either through direct application or for chemical modification. The approach we have developed of screening in combination with DNA damaging agents can be used to develop NER as a target; potentially offering a much-needed new class of antimicrobial.

## Materials and methods

### Bacterial strains, media, and culture conditions

The strains used in this study include *E. coli* MG1655, MG1655 Δ*uvrA*, MG1655 Δ*tolC*, MG1655 Δ*uvrA* Δ*tolC*, BL21 Δ*uvrA* Δ*uvrB* and EC958 [35]. The knockout strains were previously generated [19] using P1 transduction of the respective gene deletions from the Keio collection [48]. Luria Bertani broth and agar was used for primary culture and maintenance of bacterial strains. Strains were sub-cultured in MOPS (Melford, Berkshire, UK) minimal media (pH 7.4) [49] supplemented with 0.4% glucose, 1.32 mM $K_2HPO_4$ and 0.1 μg/ml thiamine. All strains were grown at 37°C with vigorous shaking.

### Antimicrobials and chemicals

The library of 2731 FDA-approved drugs were purchased from MedChemExpress, cisplatin and 9-aminoacridine were purchased from Sigma, Dorset, UK. All other compounds used for testing antimicrobial activity were purchased from MedChemExpress.

### Cloning, expression and purification of *E. coli* UvrA, UvrB and UvrC

Unlabelled UvrB and UvrC (amplified from *E coli* MG1655, NC_0009313.3) were cloned into the IPTG inducible vector pJB using primer pairs—UvrB.gibson_F (5'-ATGAGATCCTCT-CATAGTTAATTTC-3') and UvrB.gibson_R (5'-GGCGCGCCTTCAGGTAGC-3') for UvrB and UvrC.gibson_F (5'-ATGAGATCCTCTCATAGTTAATTTC-3') and UvrC.gibson_R (5'-GGCGCGCCTTCAGGTAGC-3') for UvrC. Proteins were engineered with a flexible C-terminal linker, followed by an AviTag, TEV protease site and 6X His-tag for purification using a

Ni-NTA column. The unlabelled and mNeonGreen-tagged UvrA used in this study were expressed, purified and stored as described previously [19], with the exception that the salt concentration was increased to 100 mM KCl and 400 mM KCl for the storage of UvrB and UvrC, respectively.

## High-throughput screening

The FDA-approved library of 2731 compounds were screened against MG1655 and MG1655 ΔtolC in the presence of 4 μg/mL cisplatin in MOPS minimal medium [49] as described in Fig 1A. The compounds were screened at a final concentration of 20 μM and resazurin was used to determine growth inhibition (Fig 1B). After initial screening, compounds exhibiting growth inhibition in the presence of cisplatin were screened in the presence of UV at 75 J/m$^2$ at 254 nm using a UV cross-linker (UVP/Analytik Jena UV Crosslinker CX-2000). Compounds which retained antibacterial activity in the presence of 75 J/m$^2$ UV were moved forwards and those that did not show synergy in the presence of UV irradiation were kept aside for future characterisation. The shortlisted compounds were subsequently characterised as described in Fig 1A.

## Antibacterial susceptibility testing

The minimum inhibitory concentration of antimicrobial compounds used in this study was determined adapting CLSI guidelines [50] against MG1655, MG1655 ΔuvrA, MG1655 ΔtolC, and MG1655 ΔuvrA ΔtolC, in MOPS minimal medium [49]. Resazurin was used to determine the MIC as described previously [19].

## Checkerboard assay

The checkerboard assay was used to assess activity of test compounds in the presence of cisplatin. The broth microdilution-based checkerboard assay was used with a few modifications [51,52]. A 2-fold series dilution of drugs in DMSO were plated across the plates ([DMSO] = 2.5%). Subsequently a 2-fold dilution of cisplatin in 0.9% v/v saline was plated (2.5 μl) in the microplate carefully without mixing the two solutions. Finally, 95 μL of a bacterial suspension in MOPS minimal medium prepared adapting CLSI guidelines was added to the wells [49,50] to ensure activity of cisplatin [14]. For the calculation of the synergy the FIC index formula was used: $FICi = FIC_A + FIC_B$ where $FIC_A = MIC_A$ combination / $MIC_A$ alone and $FIC_B = MIC_B$ combination / $MIC_B$ [34].

## NADH-linked ATPase assay

This assay was modified from the previously described protocol [19] to enable larger numbers of compounds to be assessed in a 96-well plate format. A 50 nM UvrA master mix (50 mM Tris (pH 7.5), 50 mM KCl, 10 mM MgCl$_2$, 0.5 mM phosphoenolpyruvate, 1 mM DTT, 210 μM NADH, 2% v/v pyruvate kinase (600–1000 U/ml) and lactate dehydrogenase (900–1400 U/ml, premixed stock from Merck)) was prepared for each independent replicate. Each reaction well consisted of a 200 μL of the above master mix and 20 μM of drug. In addition, an untreated control containing 200 μL 50 nM UvrA master mix and 2.5% DMSO were performed per plate, this yielded the basal activity of UvrA. The components were incubated at room temperature for ~5 minutes and then the reaction was started with the addition of 1 mM ATP. After ~6 minutes 0.1 ng/μl pUC18 DNA was added to the reaction mix. All results were reported as a percentage of this basal activity. The experiments were performed in triplicate and the error bars represent the standard error of the mean.

## Agarose gel-based DNA incision assay

Each incision reaction was carried out in 20 μL final volume of ABC Buffer (50 mM Tris (pH 7.5), 50 mM KCl, 10 mM MgCl$_2$, 0.0001% sodium azide) containing 5 mM DTT, 1 mM ATP, 100 nM UvrA, 200 nM UvrB, 100 nM UvrC, 2.8 nM of pUC18 DNA irradiated at 200 J/m$^2$ at 254 nm in the UV crosslinker (or undamaged DNA), and 20 μM of compound when indicated. The mixtures were incubated at 37˚C for 30 mins and the reaction was stopped by heat inactivation at 65˚C for 10 mins. DNA was separated and visualised on a 0.8% agarose gel.

## Fluorescence-based incision assay

Two complementary oligonucleotides were designed to incorporate a fluorescein modified thymine as a substrate for NER [53]. /BHQ/GT AAC TAA GCT TGA CGA TGG AGC CGT AAC AGT ACG TAG TCT G and CAG ACT ACG TAC TGT TAC GGC TCC ATC /**F**T/TC AAG CTT AGT TAC /Cy5/ (Integrated DNA Technologies, IDT, UK). At the 3' terminus of the damage (fluorescein) containing strand we placed a Cy5 fluorophore, and the 5' terminus of the complementary strand was labelled with an Iowa black hole quencher (BHQ) to quench Cy5 fluorescence when annealed. UvrC incision of the fluorescein containing oligonucleotide (14 bases on 3'and 27 bases on 5') would produce a 10-base long oligonucleotide with a melting temperature of approximately 16˚C. Since the experiment is performed at 37˚C the 10-based oligo will detach relieving the fluorophore from the quencher and producing an increase in fluorescence. The total volume per reaction in each well of a 384-well fluorescence plate was 25 μl. Each reaction mix contained 5 mM DTT, 200 nM UvrA, 400 nM UvrB, 200 nM UvrC, and 1 mM ATP in ABC buffer and was prepared on ice. 100 nM of the DNA substrate, pre-annealed by heating to 95˚C for 5 minutes and then slow cooling to room temperature was added to initiate the reaction. Cy5 fluorescence readings were taken from the bottom after 0.5/1.5/2.5/16 h at 37˚C. The outer wells were filled with water to generate a humid environment in the plate to minimize evaporation. The experiments were repeated at least twice with 2 technical replicates each, the error was reported as the standard error of the mean (n≥4).

## Repair assay for a UV damaged plasmid

Chemically competent cells of MG1655 Δ*tolC* and MG1655 Δ*uvrA* Δ*tolC* were prepared according to the method previously described [54] and stored at -80˚C. Prior to transformation 50 μl of cells were thawed on ice for 20 minutes before 100 ng of pUC18 DNA, irradiated at 200 J/m$^2$ at 254 nm, was added and incubated on ice for 30 minutes. Subsequently, cells were incubated at 42˚C for 30 seconds and then immediately placed on ice for two minutes. Cells were then transferred to 350 μl LB broth containing 2x MIC of selected antimicrobial agents and incubated for an hour at 37˚C with aeration. Cells were pelleted and resuspended in pre-warmed 350 μl LB. Aliquots where then plated in LB agar containing ampicillin (100 μg/mL) and incubated at 37˚C overnight.

To ensure that growth inhibition in this repair assay was due to activity against NER instead of the compounds directly killing the bacteria, an additional control was performed. Cells were grown to 0.5 (OD$_{600}$) from an overnight culture and after pelleting were resuspended in LB to a volume commensurate with the same cell concentration present in the UV damage repair assay. These cells were incubated for one hour in presence or absence of drug and at the end of the incubation period the bacteria were pelleted and resuspended in fresh LB broth. 5 μl were then streaked onto an LB agar plate and incubated at 37˚C overnight. In parallel, 100 μl of the cell suspension was transferred into a microtiter plate for OD$_{600}$ measurement.

## *In silico* drug docking

To study possible binding of the selected compounds to UvrA computational docking of UvrA was performed following the adapted protocol described previously [19]. Since no *E. coli* UvrA structure is available *E. coli* UvrA's protein structure was retrieved from the AlphaFold database [55,56], and then converted into pdbqt using AutoDock tools 1.5.7 [57]. The 3D structures of the compounds were downloaded from PubChem or Zinc databases [58,59], energy minimized, and converted into the pdbqt format using OpenBabel [60]. AutoDock Vina [30] was used to explore the possible binding of the most interesting compounds to UvrA. The search space was maximised to include the entirety of the protein and the exhaustiveness increased 1000 times (to 8000 from the standard setting of 8) to minimize the effect of a large search space and thus obtaining more accurate docking. As a control, ATP was docked and found to bind the distal and proximal ATPase cassettes validating the ability of the algorithm to find reliable results. Both the docking models and the binding energy were analysed to find possible binding sites on the protein and hypothesize a mechanism of action.

## Single molecule microscopy

To directly visualize protein binding to DNA we used optical tweezers coupled with fluorescence imaging (C-trap, Lumicks, NL). This system uses microfluidics to allow the capture of a single end-biotinylated Lambda DNA molecule between two streptavidin-coated silica beads. To visualise UvrA-mNeonGreen binding to DNA we transformed a plasmid containing the C-terminally tagged UvrA-mNeonGreen into BL21 Δ*uvrA* Δ*uvrB* [19] and grew the cells to mid-log phase (0.4–0.6 OD600) before induction of expression with 0.5 mM IPTG at 37°C for 3 hours. Cells were spun at 20000 rpm for 30 minutes at 4°C and resuspended in buffer (50 mM $NaH_2PO_4$, 500 mM NaCl, 15 mM imidazole pH 8). 100 μg/ml lysozyme was used with sonication to ensure complete cell lysis in the presence of protease inhibitor cocktail (no EDTA) (ThermoFisher) and 1mM PMSF. The cell debris were spun from solution at 20000 rpm for 30 mins at 4°C, and the concentration of UvrA-mNeonGreen in the supernatant was determined by absorption at 506 nm (extinction coefficient = 116000 $M^{-1}cm^{-1}$). Prior to imaging, the lysate was diluted in ABC buffer supplemented with 5 mM DTT and 1 mM ATP to a final UvrA-mNeonGreen concentration of 5nM. Finally, the solution was clarified using a 0.22 μm syringe filter before being applied to the system.

Using the microfluidics capacity of the C-trap, we recorded UvrA-mNeonGreen binding to DNA in a channel with no compound present (untreated), followed by measuring binding to the same strand of DNA in the compound-containing channel (treated), and vice versa (n = 6 strands total). Each video was recorded for 10 minutes with a 200 ms exposure at a framerate of 2 Hz using exposure synchronisation. Videos were analysed using the TrackMate plugin of Fiji (ImageJ), to objectively count the number of binders.

## Supporting information

**S1 Table. Calculated binding energies of compounds to UvrA's ATP binding pockets and allosteric sites.**
(DOCX)

**S1 Fig. UV damage repair assay controls.**
(DOCX)

**S2 Fig. Compounds alone do not impair growth in the UV growth repair assay.**
(DOCX)

**S3 Fig. Known intercalators cannot inhibit the incision mediated by NER.**
(DOCX)

**S4 Fig. Cisplatin MIC in the presence and absence of 2.5% DMSO in EC958.**
(DOCX)

**S5 Fig. Inhibitory activity of selected hits against MG1655 in the presence and absence of 4-NQO.**
(DOCX)

## Acknowledgments

We would like to thank the Kad lab for discussions, and Dr James Leech for the initial development of the UV damage repair assay. We also thank Dr Gary Robinson (University of Kent) for helpful discussions and Dr Mark Shepherd (University of Kent) for generously providing the EC958 strain.

## Author Contributions

**Conceptualization:** Neil M. Kad.

**Data curation:** Arya Gupta.

**Formal analysis:** Lorenzo Bernacchia, Arya Gupta, Antoine Paris, Neil M. Kad.

**Funding acquisition:** Neil M. Kad.

**Investigation:** Lorenzo Bernacchia, Arya Gupta, Antoine Paris, Neil M. Kad.

**Methodology:** Lorenzo Bernacchia, Arya Gupta, Antoine Paris, Alexandra A. Moores, Neil M. Kad.

**Project administration:** Neil M. Kad.

**Resources:** Alexandra A. Moores.

**Supervision:** Neil M. Kad.

**Writing – original draft:** Lorenzo Bernacchia, Arya Gupta, Neil M. Kad.

**Writing – review & editing:** Lorenzo Bernacchia, Neil M. Kad.

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
