## [Decision Letter · Decision Letter 0]

30 Aug 2023

Dear Professor Kad,

Thank you very much for submitting your manuscript "Developing novel antimicrobials by combining cancer chemotherapeutics with bacterial DNA repair inhibitors" for consideration at PLOS Pathogens. As with all papers reviewed by the journal, your manuscript was reviewed by members of the editorial board and by several independent reviewers. In light of the reviews (below this email), we would like to invite the resubmission of a significantly-revised version that takes into account the reviewers' comments.

I am returning your manuscript with three reviews. The reviewers have reached different conclusions about the paper, as you will see. Having read the assessments and reviewed the manuscript, I recommend Major Revisions and do not reject your manuscript as recommended by reviewer #2. Indeed, despite the present form need to address the issues raised by the reviewers, I feel that your study is interesting for tackling antimicrobial resistance by providing new targets. I am sorry I cannot be more positive at the moment, but look forward to receiving your revision. Note that we may refer your manuscript to some of the more critical reviewers upon resubmission.

I would like to draw your attention to the issue that I considered critical and I expect answers +/- new data. For the other, you are free to decide if the comments require new data or new explanation arguing in support with your claiming or against the reviewers’ comments.

As mentioned by reviewer #1, the novelty of this manuscript lies in targeting bacterial UvrA that is structurally different from those in Human. Although, the authors present evidence that these drugs inhibit UvrA (E. coli), the first critical issue, as mentioned by reviewer #2, is the lack of results showing that human UvrA is unaffected by these drugs.

The second critical issue is that neither cisplatin nor UV radiation are specific to NER, as they also impact replication and recombination in E. coli. Therefore, given that the aim of this manuscript is to present the first steps towards the innovative idea of anti-NER treatments, it is pertinent to clarify this point. I found reviewer #2's proposal interesting to consider as an interesting complementary experiment to strengthen the data.

The third critical issue is the lack of results arguing in support the results showing that the five drugs are lethal to E. coli without any treatment, since UvrA is not essential in E. coli.

In addition to the reviewers’ comments, I would like you to pay attention to my minor comments I have listed below.

It should be of interest to discuss a little bit more the concept of anti-NER treatment as adjuvant. . From my point of view, the adjuvants aim to make a normally resistant antimicrobial (with high MIC) effective (by decreasing the MIC) against a pathogen (here E. coli). However, here, the cisplatin concentration is necessarily fixed due to its use as an anti-cancer drug. What are you expecting : (i) anti-NER drug to be more efficient to clear infection or (ii) being efficient against cisplatin- resistant bacterial infections (if they exist)?I encourage you to clarify whether you are expecting the anti-NER drugs to be bactericidal or bacteriostatic.You should consider toning down the idea that targeting the NER could offer a new therapeutic approach. Indeed, it seems a bit of a stretch. In your study, anti-NER drugs decrease cisplatin concentrations that affect E. coli growth. So this only applies to cancer patients (treated with cisplatin) susceptible to superinfection.Please, justify why E. coli is a relevant pathogen for demonstrating the effect of anti-NER drugs on cancer patients. In cisplatin-treated cancer, is this pathogen most prevalent in event of infection?Pay attention to reviewer #2's comment on figure 2C.

We cannot make any decision about publication until we have seen the revised manuscript and your response to the reviewers' comments. Your revised manuscript is also likely to be sent to reviewers for further evaluation.

Sincerely,

Thomas Guillard, PharmD, PhD

Academic Editor

PLOS Pathogens

David Skurnik

Section Editor

PLOS Pathogens

Kasturi Haldar

Editor-in-Chief

PLOS Pathogens

orcid.org/0000-0001-5065-158X

Michael Malim

Editor-in-Chief

PLOS Pathogens

orcid.org/0000-0002-7699-2064

I am returning your manuscript with three reviews. The reviewers came to different conclusions about the paper, as you will see. After reading the reviews and looking at the manuscript, I recommend Major Revision and did not reject your manuscript as recommended by the reviewer #2. Indeed, despite the present form need to address issues raised by reviewers, I have the gut feeling that your study is of interest to tackle antimicrobial resistance by providing new targets. I am sorry I cannot be more positive at the moment, however I am looking forward to receiving your revision. Note that we may send your paper back to some of the more critical reviewers upon resubmission.

I would like to draw your attention to the issue that I considered critical and I expect answers +/- new data. For the other, you are free to decide if the comments require new data or new explanation arguing in support with your claiming or against the reviewers’ comments.

As mentioned by reviewer #1, the novelty of this manuscript lies in targeting bacterial UvrA that is structurally different from those in Human. Although, the authors present evidence that these drugs inhibit UvrA (E. coli), the first critical issue, as mentioned by reviewer #2, is the lack of results showing that human UvrA is unaffected by these drugs.

The second critical issue is that neither cisplatin nor UV radiation are specific to NER, as they also impact replication and recombination in E. coli. Therefore, given that the aim of this manuscript is to present the first steps towards the innovative idea of anti-NER treatments, it is pertinent to clarify this point. I found reviewer #2's proposal interesting to consider as an interesting complementary experiment to strengthen the data.

The third critical issue is the lack of results arguing in support the results showing that the five drugs are lethal to E. coli without any treatment, since UvrA is not essential in E. coli.

In addition to the reviewers’ comments, I would like you to pay attention to my minor comments I have listed below.

- It should be of interest to discuss a little bit more the concept of anti-NER treatment as adjuvant. . From my point of view, the adjuvants aim to make a normally resistant antimicrobial (with high MIC) effective (by decreasing the MIC) against a pathogen (here E. coli). However, here, the cisplatin concentration is necessarily fixed due to its use as an anti-cancer drug. What are you expecting : (i) anti-NER drug to be more efficient to clear infection or (ii) being efficient against cisplatin- resistant bacterial infections (if they exist)?

- I encourage you to clarify whether you are expecting the anti-NER drugs to be bactericidal or bacteriostatic.

- You should consider toning down the idea that targeting the NER could offer a new therapeutic approach. Indeed, it seems a bit of a stretch. In your study, anti-NER drugs decrease cisplatin concentrations that affect E. coli growth. So this only applies to cancer patients (treated with cisplatin) susceptible to superinfection.

- Please, justify why E. coli is a relevant pathogen for demonstrating the effect of anti-NER drugs on cancer patients. In cisplatin-treated cancer, is this pathogen most prevalent in event of infection?

- Pay attention to reviewer #2's comment on figure 2C.

Reviewer's Responses to Questions

**Part I - Summary**

Reviewer #1: Bernacchia et al. present initial steps towards the innovative idea of targeting the bacterial nucleotide excision repair (NER) system in combination with chemotherapeutic treatment. Because chemotherapy typically also suppresses the patients’ immune system, fighting infection in these patients by selectively killing pathogenic bacteria is a well-established and logical approach. The novelty of this manuscript lies in addressing prokaryotic NER, specifically the UvrA enzyme of prokaryotic NER. Conveniently, prokaryotic NER involves proteins, which are structurally completely different from those in eukaryotic NER, allowing the selective targeting of the bacterial DNA repair system (not the human). The authors present an optimised approach for screening for drugs (from an FDA approved compound library) that selectively target the bacterial NER system. Using a cell growth assay, they identified drugs that fulfil the requirement of being non-toxic for E. coli cells in the absence of additional cisplatin application as well as toxic in combination with cisplatin specifically under UV irradiation conditions (since the resulting UV lesions in DNA require repair by the NER system). The authors then tested the effects of the five most effective drug compounds on NER activity using a fluorescence based, spectroscopic in vitro DNA incision assay. They further determined (likely) binding positions of the compounds on UvrA via molecular docking and used a fluorescence-based UvrA ATPase activity assay as well as single molecule visualisation of DNA binding by UvrA to characterise the effects of these five best hits.

I find this study very interesting and highly relevant (in light of growing antibiotic resistances). The manuscript is in general well written and the aim, approaches, and findings clearly presented. Nevertheless, I have a few minor issues with the manuscript as listed in detail below.

Reviewer #2: The manuscript by Bernacchia outlines an approach to address bacterial infections during chemotherapy. This is a critical issue, given that many chemotherapeutic agents suppress the immune system. While the use of cisplatin is understandable for cancer therapy, it's not ideal for inhibiting NER.

My primary concern is the specificity of the five drugs presented as NER inhibitors. Firstly, neither cisplatin nor UV radiation is specific to NER, as they also impact replication and recombination in E. coli. A drug like NQO or NFZ might offer a better starting point to demonstrate specificity.

In their in vitro experiments, the authors present evidence that these drugs inhibit UvrA. However, essential controls appear to be missing. For instance, it would be crucial to show that human UvrA is unaffected by these drugs, or at a minimum, that other proteins with ATPase activity are not inhibited. While inhibiting an activity in vitro can be straightforward, demonstrating specificity can be more challenging.

Lastly, I'm puzzled by the observation that their five drugs are lethal to E. coli even in the absence of any treatment, especially since UvrA is not essential in E. coli. This raises concerns about the specificity of the drugs in question.

Reviewer #3: I find that the manuscript presents a very important amount of encouraging findings. It is clearly written and is convincing.

**Part II – Major Issues: Key Experiments Required for Acceptance**

Reviewer #1: (No Response)

Reviewer #2: (No Response)

Reviewer #3: (No Response)

**Part III – Minor Issues: Editorial and Data Presentation Modifications**

Reviewer #1: Methods

Is it common to present Materials & Methods details only in the Supplemental section in PLOS Pathogens?

Results

1) Page 4 line 78: cisplatin concentration is given in units of μg/ml while all concentrations for the tested compounds are given in units of molarity throughout the manuscript. It might be easier for comparison to also give the cisplatin concentration in M.

2) Page 5 line 103: I think it is not entirely clear whether the five selected, best hits from the screening procedure came from the MG1655 strain or from the drug pump deficient MG1655 ΔtolC strain. It may be interesting to know whether drug efflux played a role and for which of the compounds. It is mentioned later in the manuscript (on page 10) that out of the five hits, Pirarubicin, Mitoxantrone, and 9-aminoacridine were identified from the wild type strain (with functional drug efflux pump). Two of these compounds are known to intercalate into DNA as stated by the authors, and the third (9-aminoacridine) has been shown, as mentioned by the authors in the Discussion, to modify DNA in combination with cisplatin. Is it possible that stable interactions with DNA by these three compounds are responsible for their retention in the cell in the wild type strain? Such DNA modification may also possibly prevent either binding to the DNA by UvrA or NER incision of the DNA, although the authors showed that other intercalators did not inhibit NER (Figure S3, see also point 9 below)? Is DNA binding by UvrA affected by 9-aminoacridine? The enhancement of ATPase activity in the presence of DNA seems only minor with this compound, while it is known that the ATPase activity of (wild type) UvrA is strongly enhanced in the presence of DNA (see point 7 below).

3) Page 5 line 113/114: Why would incision of a single DNA strand 10 nt from the 3’ end (with the attached fluorophore) result in “the 10 nt fragment leaving the duplex”? Is The DNA heated / what is the melting temperature?

4) Page 6 last paragraph: include mentioning of Figures S1 and S2 and possibly also mention that drugs alone did not impair bacterial growth (Figure S2), since this seems like an important information.

5) Figure 2: (B) RFU not introduced; (D) CFU not introduced – only in Supplement; (C) arrows indicating which is the incision product would help here, it appears that the upper band is the incision product not the lower (usually the slower migrating species, i.e. the full length DNA not the incised, is shown at the top and the shorter DNA below); for L-thyroxin an incubation time of 15 minutes is stated (in contrast to the 30 minutes in (B)) – are all incubation times 15 minutes for all the compounds in (C)? It is surprising that the three assays, the fluorescence-based and gel-based incision assay and the in vivo UV recovery assay, give such different results for the five tested compounds. In all cases (or in most cases) all three assays confirmed suppression of NER activity, but the degree to which they did this was not consistent between the different assays (e.g. Dienestrol almost not significant in fluorescence-based and gel-based incision assay but best in in vivo assay). What could be reasons for this?

6) Page 7 mention Figure 3B (e.g. line 159)

7) Figure 3: (A) I think it might be helpful to include UvrA without drug as a reference for the enhancement of ATPase activity by DNA (presumably the 100% is for UvrA without DNA). Also: what is the mechanism of NER inhibition by 9-aminoacridine, since ATPase activity by UvrA seems unaffected, at least in the absence of DNA. It also does not seem to bind to UvrA in the docking studies? See also point 2 above. (B) A colour legend would be helpful or, alternatively, the histogram bars in (C) could be colour-coded matching the colours of the different drugs in the zoom in (B).

8) Page 8 second paragraph: According to Figure 3B, L-Thyroxine and Dienestrol bind to allosteric sites at a small distance from the ATP binding sites in UvrA, while Mitoxantrone and Pirarubicin bind in the proximal ATP binding site of UvrA. From the ATPase assay it is not clear whether ATP binding or hydrolysis are affected for each of the five compounds. Is it only ATP hydrolysis that is reduced for Pirarubicin and Mitoxantrone or is it ATP binding per se, which may be partially blocked by the bound drugs. For the two drugs bound in the allosteric sites, L-Thyroxine and Dienestrol, single molecule fluorescence microscopy was carried out in addition, to observe effects on DNA binding by UvrA. The ATPase assay indicated that these two drugs had the strongest effect in suppressing UvrA’s ATPase activity. For Dienestriol it appeared like the enhancement of ATPase activity in the presence of DNA was still present. While the single molecule visualisation is very cool, in principle a simple gel assay (EMSA) would have probably done the job. Here, while it is very clear that in the presence of Dienestriol or L-Thyroxine DNA binding by UvrA was strongly suppressed, the exact mechanism is not clear. Would an ATPase deficient mutant of UvrA bind to the DNA similar as wild type? Is DNA binding per se (in the absence of damage) not independent of UvrA ATPase activity? The authors have shown in a previous work that without ATP hydrolysis UvrA stays bound to DNA longer (Barnett and Kad, 2019, FASEB J). Or is DNA binding blocked by the bound drugs (which the ATPase data for Dienestriol would argue against)?

9) Page 8: Figure S3 (no NER inhibition by other intercalators) should be mentioned in Results section before Figure S4 (e.g. lines 172-174).

10) The calculation of the FIC index may want to be included in the Methods section. Mention Figure 4B in the text (e.g. page 11 top)? Also, the effects of Pirarubicin, Mitoxantrone, and 9-aminoacridine, the three drugs tested on the multi-drug resistant strain EC958 as well as the MG1655 lab strain were rather different between the two strains. While they were all clearly effective, this raises the issue that their effects may vary drastically between different bacteria. Nevertheless, I believe this study is an excellent starting point for the exploration of the bacterial NER inhibition route in support of chemotherapy.

Reviewer #2: (No Response)

Reviewer #3: Following a comment from previous reviewers, I think that the checkerboard assay is sufficient to show growth inhibition. As noted by one of the reviewers, if the authors want to conclude about killing or survival, then they would need to show a time kill assay (for the most promising condition for example). However, I think the checkerboard assay would be sufficient to show synergy between cisplatin and the tested drugs.

PLOS authors have the option to publish the peer review history of their article (what does this mean?). If published, this will include your full peer review and any attached files.

Reviewer #1: **Yes: **Ingrid Tessmer

Reviewer #2: No

Reviewer #3: No
---

## [Decision Letter · Decision Letter 1]

26 Oct 2023

Dear Professor Kad,

Thank you very much for submitting your manuscript "Developing novel antimicrobials by combining cancer chemotherapeutics with bacterial DNA repair inhibitors" for consideration at PLOS Pathogens. As with all papers reviewed by the journal, your manuscript was reviewed by members of the editorial board and by several independent reviewers. The reviewers appreciated the attention to an important topic. Based on the reviews, we are likely to accept this manuscript for publication, providing that you modify the manuscript according to the review recommendations.

I am returning your manuscript with three reviews from the referees I had solicited. The reviewers came to different conclusions about the paper, as you will see. After reading the reviews and looking at the manuscript, I have decided that the only further experiments requested by reviewer 2 necessary for this manuscript to meet the criteria for publication at PLOS Pathogens is to assess whether your NER inhibitors will behave in the presence of ciprofloxacin.

There are, however, a few remaining minor revisions that need to be addressed to prepare the manuscript for publication. If all the following items are addressed, I hope to be able to make a final decision without sending the manuscript out for a second round of review.

Please pay particular attention to my following suggestions and give them due consideration.

1) Assess NER inhibitors upon exposure to ciprofloxacin to answer to reviewer#2.

2) Please tone down the specificity of your drugs towards the NER system by mentioning that UV/cisplatin-induced lesions block DNA replication.

3) Clearly state that you have not assessed whether your drugs had no impact on the human homolog of UvrA. However, (i) the closest sequence homolog (25% homology) of uvrA is an ABC transporter of insulin due to its ATPase domain (see reference provided by reviewer#3) and (ii) the drugs used here are FDA approved (I agree with reviewer#3), therefore, we can anticipate no innocuity for human and therefore no effect on potential human NER proteins or any other ATPase proteins.

4) Provide data (from literature, database…) indicating that E. coli is mainly found in infections occurring in cisplatin-treated cancer patients. Indeed, your reply to my previous comment merely explains that urosepsis is due to resistant E. coli. It is worth justifying your assay with E. coli and not another pathogen in the clinical context of the patients you are targeting.

Sincerely,

Thomas Guillard, PharmD, PhD

Academic Editor

PLOS Pathogens

David Skurnik

Section Editor

PLOS Pathogens

Kasturi Haldar

Editor-in-Chief

PLOS Pathogens

orcid.org/0000-0001-5065-158X

Michael Malim

Editor-in-Chief

PLOS Pathogens

orcid.org/0000-0002-7699-2064

I am returning your manuscript with three reviews from the referees I had solicited. The reviewers came to different conclusions about the paper, as you will see. After reading the reviews and looking at the manuscript, I have decided that the only further experiments requested by reviewer 2 necessary for this manuscript to meet the criteria for publication at PLOS Pathogens is to assess whether your NER inhibitors will behave in the presence of ciprofloxacin.

There are, however, a few remaining minor revisions that need to be addressed to prepare the manuscript for publication. If all the following items are addressed, I hope to be able to make a final decision without sending the manuscript out for a second round of review.

Please pay particular attention to my following suggestions and give them due consideration.

1) Assess NER inhibitors upon exposure to ciprofloxacin to answer to reviewer#2

2) Please tone down the specificity of your drugs towards the NER system by mentioning that UV/cisplatin-induced lesions block DNA replication

3) Clearly state that you have not assessed whether your drugs had no impact on the human homolog of UvrA. However, (i) the closest sequence homolog (25% homology) of uvrA is an ABC transporter of insulin due to its ATPase domain (see reference provided by reviewer#3) and (ii) the drugs used here are FDA approved (I agree with reviewer#3), therefore, we can anticipate no innocuity for human and therefore no effect on potential human NER proteins or any other ATPase proteins.

4) Provide data (from literature, database…) indicating that E. coli is mainly found in infections occurring in cisplatin-treated cancer patients. Indeed, your reply to my previous comment merely explains that urosepsis is due to resistant E. coli. It is worth justifying your assay with E. coli and not another pathogen in the clinical context of the patients you are targeting.

Reviewer Comments (if any, and for reference):

Reviewer's Responses to Questions

**Part I - Summary**

Reviewer #1: All of my comments have been adequately addressed by the authors.

Reviewer #2: 1. We strongly disagree with the reviewer on this point, the very origin of the protein names found in NER derives from UV resistance. NER is the primary mechanism of repairing UV damage (Boyce & Howard-Flanders, 1964; Franklin & Haseltine, 1984; Pettijohn & Hanawalt, 1964; Sancar & Rupp, 1983; Setlow & Carrier, 1964; Thomas et al., 1985) and cisplatin induced damage (Beck et al., 1985; Husain et al., 1985; Popoff et al., 1987). With regards the second point, we recently published data using 4- NQO in comparison to UV in uvra+ and uvra- E. coli strains(Bernacchia et al., 2022), these data are presented below.

I strongly disagree with the authors regarding the specificity of UV and cisplatin. Citing papers from the last century does not strengthen their case. While they are correct that UV/cisplatin lesions are removed by NER, these lesions also block DNA replication. For instance, a recF mutant is sensitive to both drugs and is not involved in NER (see Justin Courcelle, David J. Crowley, and Philip C. Hanawalt, J. Bact. 1999). Furthermore, their argument about the greater change in MIC for a UvrA knockout strain versus WT when treated with UV as opposed to 4-NQO is flawed. These are entirely different treatments, and their specificity can't be compared. It seems conducting the NQO experiments would be simple, cost-effective, and provide a clear answer. I am curious about how these new NER inhibitors will behave in the presence of Cipro, which doesn't require NER.

2. As stated earlier, humans do not possess UvrA. There is a functional conservation of NER but no structural similarities exist (Petit & Sancar, 1999). Other (but not all) DNA repair pathways in bacteria do have human counterparts with structural conservation, which is why inhibiting NER is so attractive.

I recognize that there's no structural homolog in humans. While inhibiting an activity in vitro might be straightforward, proving specificity can be more challenging. For instance, urea can inhibit almost any protein in vitro but isn't a specifically good inhibitor. What experiments have the authors conducted to demonstrate specificity in vitro? It is essential to show that these inhibitors don't inhibit the ATPase activity of unrelated proteins in my opinion.

3. There is a problem of copy and paste with fig2c western blot as I already mentioned in my previous review.

Reviewer #3: In my opinion, the authors have satisfactorily answered all comments. The use of a clinically relevant E. coli strain is a plus for this study.

Regarding UvrA homologues in human cells and the possibility of the existence of a human UvrA homologue, the NER machinery in human cells involves a multitude of proteins (about 30 factors), which act in complexes. For a review: https://www.nature.com/articles/nrm3822

None of these factors show homology with E. coli UvrA, the closest sequence homologue (25% homology) to uvrA is an ABC transporter of insulin because of its ATPase domain.

XPC is the main initiator of NER, except at UV lesions where the lesion is recognized by DDB.

Regarding DDB: “DDB may play a special role in the repair of UV damage, but it cannot be the sole damage recognition subunit of human excision nuclease” PMID: 8407968

“physical interaction between DDB and XPA plays an important role in the DDB-mediated NER reaction” PMID: 19056823

More importantly, the tested drugs have been already FDA-approved, which means that their innocuity for human cells has already been established. This is one of the advantages of drug repurposing, where drugs that are already used in humans would be used for another purpose. The combination of cisplatin and the selected drugs has not been tested yet but this would be, in my opinion, more relevant for clinical studies, rather than the proof of concept of potentiation of antibmicrobial action of the combination, as shown here.

**Part II – Major Issues: Key Experiments Required for Acceptance**

Reviewer #1: (No Response)

Reviewer #2: See previous section

Reviewer #3: (No Response)

**Part III – Minor Issues: Editorial and Data Presentation Modifications**

Reviewer #1: (No Response)

Reviewer #2: (No Response)

Reviewer #3: (No Response)

PLOS authors have the option to publish the peer review history of their article (what does this mean?). If published, this will include your full peer review and any attached files.

Reviewer #1: **Yes: **Ingrid Tessmer

Reviewer #2: No

Reviewer #3: No

Figure Files:

Data Requirements:

Reproducibility:

References:

---

## [Editor Report · Decision Letter 2]

28 Nov 2023

Dear Professor Kad,

We are pleased to inform you that your manuscript 'Developing novel antimicrobials by combining cancer chemotherapeutics with bacterial DNA repair inhibitors' has been provisionally accepted for publication in PLOS Pathogens.

Best regards,

David Skurnik, M.D., Ph.D.

Section Editor

PLOS Pathogens

Kasturi Haldar

Editor-in-Chief

PLOS Pathogens

orcid.org/0000-0001-5065-158X

Michael Malim

Editor-in-Chief

PLOS Pathogens

orcid.org/0000-0002-7699-2064
---

## [Editor Report · Acceptance letter]

2 Dec 2023

Dear Professor Kad,

We are delighted to inform you that your manuscript, "Developing novel antimicrobials by combining cancer chemotherapeutics with bacterial DNA repair inhibitors," has been formally accepted for publication in PLOS Pathogens.

Best regards,

Michael Malim

Editor-in-Chief

PLOS Pathogens

orcid.org/0000-0002-7699-2064